# OpenReview forum: "Beyond Jailbreaking: Auditing Contextual Privacy in LLM Agents"
_ICLR.cc/2026/Conference — Submitted to ICLR 2026_

### Official Review · Reviewer_WcPM · 2025-10-30

**Soundness:** 2
**Presentation:** 1
**Contribution:** 2
**Rating:** 2
**Confidence:** 3

**Summary:**

The paper introduces a framework, Conversational Manipulation for Privacy Leakage (CMPL), to audit contextual privacy risks in LLM agents. It models a multi-turn interaction between an application agent, an adversarial agent, and an auditor that detects both explicit and implicit leakages. Experiments are conducted across multiple scenarios (insurance claims, interview scheduling, etc.), demonstrating that adaptive multi-turn adversaries can induce significant privacy leakages even when single-turn baselines fail. The paper positions CMPL as both an auditing framework and an open benchmark for studying conversational privacy.

**Strengths:**

1. The paper studies privacy in multi-turn, context-rich conversational settings, which is an important area as LLM agents are increasingly used in personal and enterprise contexts.
2. The experiments are comprehensive across multiple domains, adversarial strategies, and model configurations, and the data artifact may be useful for the community.

**Weaknesses:**

1. Despite the paper’s comprehensiveness, the core contribution is not sufficiently clear. The metrics for “contextual privacy leakage” (explicit vs. implicit) are heuristic and somewhat circularly defined based on the chosen adversarial model and auditor. For example, there are many metrics in main result figures. While this make the analysis comprehensive, it's unclear how to use them in practice when we need to draw a conclusion on the privacy robustness of a certain model or agent. The proposed “risk quantification” lacks theoretical grounding or interpretability beyond specific instantiations, either.
2. The length and technical density of the paper are not commensurate with its conceptual novelty. Much of the design (multi-turn adversary, auditor-based evaluation, LLM-as-judge) builds on existing frameworks such as red teaming and contextual integrity audits, with relatively limited new methodological insight.
3. The evaluation is heavily conditioned on a specific adversarial agent implementation. It remains unclear whether the observed “zero success rate” or high success rate from this particular adversary meaningfully implies privacy safety or vulnerability in general. The conclusions therefore risk being artifact-dependent.

**Questions:**

1. Do you think the metrics in the paper can be simplified? For example, can we separate them into main metrics and auxiliary metrics?
2. The study is conducted using 5 specific scenarios as the testbed. Can the proposed CMPL metrics be used as quantitative auditing standards, or are they primarily diagnostic tools within this specific setup?

---

> ### Author Response · Authors · 2025-11-20
> **Rebuttal to Reviewer WcPM (Part 1 of 2)**
>
> We thank Reviewer WcPM for their feedback. We address the main concerns regarding novelty, metrics, and generalizability below.
> We also encourage the reviewer to check the other reviewer's assessment and the contributions of our work, as we found some disconnect with respect to, for instance reviewers yRR8 and UWMf, who found our contribution "excellent" and "substantial."
>
> **1. On "Limited New Methodological Insight" and "Unclear Contribution"**
>
> Our central contribution is the **CMPL auditing framework itself, the first systematic auditing scheme for contextual integrity in multi-turn, stateful conversational agents.** To our knowledge, no prior work has introduced such an audit for this specific setting. We invite you to also please see other reviewers assessment regarding this aspect. We believe our framework is a significant methodological advance, and has potential to spur further research in the agentic contextual privacy field. It is accompanied by three key novel components:
>
> 1.  **A Novel Adversary Class:** As a key part of our audit, we are the first to formalize an **adaptive, planning-based adversary** that uses distinct **Strategist ($S_\Theta$)** and **Prompt Generator ($G_\Theta$)** modules. This is not a simple "red team" script; it's an agent that *reasons* about the conversation and its privacy ramifications, plans multiple steps ahead (sub-goals), and *adapts* its strategy. Its power is proven by its ability to achieve high ASR (up to 75%) even where SOTA baselines [Bagdasaryan et al., 2024, Abdelnabi et al., 2025] achieve 0% ASR.
> 2.  **A Novel Threat Vector (Implicit Leakage):** Our framework is the first to formally define, measure, and audit **implicit leakage** (side-channel attacks). Our Side-channel Predictor ($P_\Theta$) based on bootstrap confidence and the analysis of the adversary's posterior belief evolution (Section 4.2) are entirely novel. This demonstrates a critical class of threats, where a secret is inferred without ever being explicitly stated, that prior audits, which focus on explicit disclosures, would completely miss.
> 3.  **Empirical Insights:** Our audit reveals that *even when the adversary fails to predict the correct value*, it may still learn significant information, as shown by large spikes in its posterior belief (Section 4.2). This is surprising, and novel, to our knowledge.
>
> **2. On Metrics: "Heuristic," "Circular," and "Artifact-Dependent"**
>
> We believe that this stems from a slight misunderstanding  about the term *audit*. In addition, it is not elaborated how our metrics (most of which are popular metrics used in prior work) are "circular":
>
> * First, an audit, by definition, is an *empirical evaluation* of a specific artifact (the agent) against another specific artifact (the adversary). As explicitly noted by prominent work in privacy auditing (including [Steinke et al., NeurIPS 2023 Outstanding Paper]), a formal proof provides a theoretical *upper* bound on privacy, whereas **"a privacy audit provides an empirical *lower* bound"**. Security and privacy literature tends to proceed as an arms race of improving attacks and defenses, thus such audits capture vulnerability to the strongest artifact developed thus far. We show that agents secure against *old* artifacts (baselines [Bagdasaryan et al., 2024, Abdelnabi et al., 2025]) are *insecure* against our *new, more powerful* artifact (the CMPL adversary), an evaluation enabled by our auditing framework (which can be used for any adversary). This empirical audit methodology is the accepted standard, well-precedented in LLM benchmarks/safety audits [Chao et al., 2024, Andriushchenko et al, 2025]. In addition, our chief contribution, *the auditing framework, is general and can also be used to audit vulnerability to any future adversary*.
> * Second, regarding generalizability: We have demonstrated that our findings hold across **three diverse scenarios** (Insurance, Scheduling, and Credit Card Application) and across multiple model families. Furthermore, we show clear **scaling laws** (Table 1), where attack success scales with model size (7B to 70B). Furthermore, the proposed adversaries consistently outperform baseline adversaries across the choice of scenarios and targets. This proves our findings are a generalizable property of the methodology, not an artifact of the scenarios or the choice of adversary.
>
> (Response continued below.)

---

> ### Author Response · Authors · 2025-11-20
> **Rebuttal to Reviewer WcPM (Part 2 of 2)**
>
> * **In response to Q1 (simplifying metrics):**
> We respectfully disagree that our metrics are unnecessarily complex. On the contrary, we believe the set we use is already a minimal and essential suite, clearly defined in Section 3. Our **main metric, ASR**, captures the efficacy of the attack as an empirical lower bound on privacy risk and is a standard choice in prior work on attacks and auditing. However, an audit that reports only ASR would be incomplete, since it would ignore how and when leakage occurs and whether it can happen without explicit disclosure.
> \
> For this reason, our **auxiliary metrics** are necessary rather than optional. **PLTC** quantifies *stealth* by measuring whether the attack succeeds before the task is completed (c.f. similar agent conversation exit notion in [Zhou et al, 2024]). **Temporal dynamics** ($\gamma_t$) characterize *when* the agent is most vulnerable over the course of the interaction. **Belief updates** quantify *information gain* about the secret, even in the absence of explicit leakage. Providing an empirical basis for conclusions about a model’s privacy robustness is precisely the goal of a privacy auditing exercise, and this requires more than a single scalar. Our metric suite is designed to meet exactly this goal.
> \
> Finally, we emphasize that these metrics, in particular **ASR** and belief-based measures ([Chao et al, 2024], [Russinovich et al, 2025], [Dwork et al, 2006], [Kasiviswanathan et al, 2014], [Nanayakkara et al, 2023]), are *standard and widely used* in the privacy and security literature, and are not specific or unique to our work.
>
> * **In response to Q2 (are metrics a general standard?):** Yes. The proposed CMPL auditing tools and methods are designed to serve as quantitative auditing standards beyond the specific scenarios considered in this work. Indeed, the main contribution of our paper is the auditing framework itself, while the proposed adversaries and scenarios simply provide concrete settings for demonstrating the utility of the auditors. Put differently, *the design of the auditor and associated analyses/metrics are independent of the choice of scenario and can be extended to any arbitrary setting.*
>
> **3. On Presentation**
>
> We were surprised by the ''1: poor'' presentation score, as it contrasts with the other reviews. Two other reviewers rated the presentation as ''3: good,'' and Reviewer yRR8 explicitly described the paper as ''clearly and effectively written.'' Could you please clarify which aspects of the presentation you found to be poor (for example, organization, notation, or clarity of the experimental setup)? This would be extremely valuable to us in further improving the manuscript.
>
> ---
>
> We sincerely appreciate the time and care you have invested in reviewing our work and, in light of the clarifications and new results provided in this rebuttal, we would be very grateful if you would consider updating your scores accordingly.
>
> ---
>
> ## References
>
> [Chao et al, 2024] Chao, Patrick, et al. "JailbreakBench: An Open Robustness Benchmark for Jailbreaking Large Language Models." NeurIPS Datasets and Benchmarks Track, 2024.
>
> [Andriushchenko et al, 2025] Andriushchenko, Maksym, et al. "AgentHarm: A Benchmark for Measuring Harmfulness of LLM Agents." ICLR, 2025.
>
> [Steinke et al, 2023] Steinke, Thomas, et al. “Privacy Auditing with One (1) Training Run.” Advances in Neural Information Processing Systems, 2023
>
> [Russinovich et al, 2025] Russinovich, Mark et al. “Great, Now Write an Article About That: The Crescendo Multi-Turn LLM Jailbreak Attack.”, Proceedings of the 34th USENIX Conference on Security Symposium, 2025
>
> [Dwork et al, 2006] Dwork, C., McSherry, F., Nissim, K., Smith, A. (2006). Calibrating Noise to Sensitivity in Private Data Analysis. In: Halevi, S., Rabin, T. (eds) Theory of Cryptography. TCC 2006. Lecture Notes in Computer Science, vol 3876. Springer, Berlin, Heidelberg. https://doi.org/10.1007/11681878_14
>
> [Kasiviswanathan et al, 2014] Kasiviswanathan, Shiva P., and Adam Smith. 2014. “On the ’Semantics’ of Differential Privacy: A Bayesian Formulation”. Journal of Privacy and Confidentiality 6 (1). https://doi.org/10.29012/jpc.v6i1.634.
>
> [Nanayakkara et al, 2023] Nanayakkara, Priyanka et al. “What Are the Chances? Explaining the Epsilon Parameter in Differential Privacy.” USENIX Security Symposium, 2023
>
> [Zhou et al, 2024] Zhou, Xuhui et al. “HAICOSYSTEM: An Ecosystem for Sandboxing Safety Risks in Human-AI Interactions.”, Conference on Language Modeling (COLM), 2025

---

> > ### Comment · Reviewer_WcPM · 2025-11-24
> >
> > I thank the authors for the detailed response and have carefully read other reviewers' review. Given that some of the reviews are very concise, even though I agree with the strengths they list, I do not think they directly anser my questions.
> >
> > **On metrics**:
> > - Could you give some examples on how temporal dynamics ($\gamma_t$) metrics can be used to interpret the adversary?
> > - Could you give some examples on how auxiliary metrics help surface nuances that ASR alone is unable to surface?
> >
> > This will provide intuitive insights on these metrics and will help the readers to better understand the table.
> >
> > **On presentation and unclear contributions**:
> > - I give a low rating to the presentation score because I think the presentation fails to deliver the main contributions in a clear way and many notations over-complicate the paper. Personally, I think emperical papers need to use math notations with caution. For [Steinke et al., NeurIPS 2023 Outstanding Paper], they need to heavily use math notations because of the definition of DP and the choice of data points are based on statistical estimation. But for this paper, given different components are instantiated by prompting LLM, why do we really need these notations?
> > - I agree with "Our framework is the first to formally define, measure, and audit implicit leakage (side-channel attacks)" and think this the core part of the auditing framework. Shouldn't we describe Section 2.3 first if this is the case? Based on my understanding of audit (as the authors also kindly provide the definition here, "an audit, by definition, is an empirical evaluation of a specific artifact (the agent) against another specific artifact (the adversary)", we need to first define what is the auditing is doing here. Then to highlight your design of the adversary is valuable and necessary, you can compare the auditing results w/ and w/o the adversary design. In this way, the readers can better understand (1) what this framework tries to audit and (2) why the design of the adversary is necessary.

---

> ### Author Response · Authors · 2025-11-25
> **Response to Official Comment by Reviewer WcPM (Part 1 of 3)**
>
> We thank the reviewer for the follow-up and for carefully reading both our rebuttal and the other reviews. We address the two groups of questions in turn.
>
> **1. On temporal dynamics and auxiliary metrics**
>
> The goal of the temporal and auxiliary metrics is to turn raw transcripts into signals that help practitioners diagnose *how* and *when* leakage occurs, not just whether the final attack succeeds.
> In our revision, we have already made the exposition more intuitive and clarified this. In particular, there are two key observations:
>
> * **How temporal dynamics help interpret the adversary**
> *
>    The temporal dynamics metrics summarize, as a function of turn index, the probability of success conditioned on having reached that turn. This allows us to answer questions such as:
>
>    > *“At which stage of the interaction is the agent most vulnerable, and how does that depend on the adversary design?”*
>
>    For example, across the scenarios we observe two qualitatively different patterns:
>
>    * In some settings, almost all successful attacks occur only in the last few turns, after the task is essentially completed. The temporal curves show that the adversary’s posterior over the true secret remains close to its prior until late in the dialogue, then sharply increases once the agent starts providing clarifying details. In such a case, a practitioner might decide that a simple mitigation, such as truncating or filtering the last turns or resetting memory after task completion, is effective.
>    * In other settings, vulnerability peaks in the *early* turns. The temporal curves show that the adversary’s belief already concentrates on the true secret after a small number of apparently innocuous questions, even though ASR is measured only at the final guess. Here, the mitigation needs to focus on early turn policies and on preventing the agent from volunteering background information that can be combined with external knowledge.
>
>    These temporal profiles also help to interpret and compare different adversary designs. As discussed in Section 4.1, the longer horizon subgoals based adversary and the immediately reactive adversary exhibit distinct behaviors in their temporal leakage curves $\gamma_t$. For the subgoals based adversary, $\gamma_t$ initially increases and typically peaks around 10–20 turns, reflecting that it first spends several turns building up context before querying for highly revealing information, and then becomes less creative thereafter. In contrast, the reactive adversary starts with a high $\gamma_t$ that gradually decreases over turns, consistent with its immediately adaptive strategy that attempts aggressive probing early and then exhausts effective tactics. This kind of analysis is only possible with temporal metrics and gives practitioners concrete intuition about how different adversaries interact with the same agent, and where in the conversation each adversary is most effective.
>
>    Note that if we only reported ASR, these two situations would appear identical. Temporal dynamics quantify, in a reproducible way, where the weak points of the conversation are and how the planning components of the adversary exploit them. In the revision we will add explicit textual examples of these patterns and a short paragraph on how to read the temporal plots.
>
>    Additionally, we observe that in several scenarios the privacy leakage likelihood initially increases and then drops after a number of turns. This drop indicates that the adversary’s intent and tactics eventually become salient to the application agent, which then becomes more effective at avoiding leakage. This insight directly suggests a class of temporal mitigation strategies, which we now validate empirically.
>
>    In Section 4.5 we add new results on a simple temporal guardrail that conditions the application agent on a longer simulated conversation history. In the insurance claim scenario (targeting family medical history), providing the agent with 40 turns of simulated prior interaction reduces ASR from 65 percent to 25 percent:
>
>    |                                | ASR |
>    | ------------------------------ | :-: |
>    | **Without temporal guardrail** |  65 |
>    | **With temporal guardrail**    |  25 |
>
>    This reduction is obtained without changing the underlying model or the adversary. It is instead a direct consequence of insights derived from the temporal dynamics analysis, which reveal when and how the agent learns to recognize adversarial behavior. We will highlight this connection more clearly in the revised manuscript so that readers can see how temporal metrics lead to actionable mitigation strategies.
>
> (Response continued below.)

---

> ### Author Response · Authors · 2025-11-25
> **Response to Official Comment by Reviewer WcPM (Part 2 of 3)**
>
> **1. On temporal dynamics and auxiliary metrics (Continued)**
>
> * **How auxiliary metrics surface nuances that ASR alone cannot**
>
>    While ASR is the primary scalar metric, the auxiliary metrics are there to resolve cases where ASR is the same but the *risk profile* is very different:
>    * **PLTC (leakage before task completion).** Consider two agents with identical ASR. In one case, successful attacks almost always occur after the user’s task has been fully completed and the user has already received a useful answer. In the other case, a nontrivial fraction of attacks succeed *before* task completion, so the user never even obtains the utility they asked for. Both agents have the same ASR, but the second one is substantially more problematic in practice. PLTC makes this difference visible by quantifying how often the adversary extracts the secret “stealthily” while the user still believes the agent is working on their task.
>    * **Belief-based information gain.** We also see settings where the adversary’s *final* guess is incorrect (ASR equal to zero or very low), yet its posterior probability on the true secret has increased sharply over the course of the conversation. In such cases, a privacy officer might consider the system unacceptable, even though ASR alone would suggest “no problem”. The belief-based metrics capture this “partial but significant” leakage and align with the Bayesian view of privacy risk used in prior work on privacy auditing and differential privacy semantics.
>    * **Interaction with adversary design.** Finally, the auxiliary metrics help distinguish adversarial strategies that have similar ASR but use very different interaction styles. For instance, an adversary that leaks only through overt, high-pressure queries is easier to detect and filter than one that gradually accumulates side-channel evidence and only makes a guess once the posterior is already highly concentrated. Our metrics separate these cases and show that the planning-based adversary we introduce tends to produce earlier and more stealthy leakage patterns than simple single-shot or static prompt baselines.
>
>    We have now explicitly added a section (Appendix F.5 with a main text reference to it in Section 3) in our revised version where we we explicitly walk the reader through these comparisons, so that the role of each auxiliary metric is immediately clear. Thank you for the suggestion; we believe it improves the clarity of the paper and helps readers further appreciate the empirical insights offered in it.
>
> **2. On presentation, notation, and the structure of contributions**
>
> We appreciate the clarification regarding your presentation score. It helps us see that your main concern is how the contribution is *communicated* rather than a disagreement about what it is.
> - **Why we use formal notation despite LLM-instantiated components**
>    Regarding notation, we appreciate the opportunity to clarify our intent and its role in the paper. Our goal was to retain only the minimal set of formalism needed to define the audit precisely and make the framework reusable. In the current draft, the notation is serving a small number of specific purposes:
>     1. To describe the structure of the application agent’s context, including conversation history and privacy norms. This is necessary to specify what information is available to the agent and adversary at each turn, and to distinguish secrets from general context.
>     2. To formalize the adversary’s belief updates (Eq. 1), which are later visualized empirically in Figure 3 (right). The adversary’s objective is to gain information about the target attribute. An increase in its posterior belief on the true value, even if it does not end with a correct explicit guess, is precisely what we mean by implicit privacy leakage. For example, if after the conversation the adversary becomes much more confident that a patient’s family has a history of cancer, this is a leakage event even if the secret is never stated verbatim.
>     3. To specify the concrete implementation of the implicit leakage auditor and adversary (Eq. 2) and the conditions under which leakage is flagged. These definitions are what allow others to reproduce and extend our auditor rather than treating it as a black box tied to one particular prompt set.
>     4. To define the soundness and completeness properties of the auditor (Eqs. 3 and 4), which in turn support the notions of soundness error, completeness error, and expected detection delay that we study empirically in Table 2. Without these definitions, it would be difficult to reason rigorously about when the auditor is trustworthy.
>
> (Response continued below)

---

> ### Author Response · Authors · 2025-11-25
> **Response to Official Comment by Reviewer WcPM (Part 3 of 3)**
>
> **2. On presentation, notation, and the structure of contributions (Continued)**
> - **Why we use formal notation despite LLM-instantiated components (continued from above comment)**
>
> We do not view these notational elements as decorative. They are intended to be the essential scaffolding that makes the metrics and the functioning of the framework unambiguous. Reviewer UWMf explicitly cited the formalization as a strength of the paper, and there is clear precedent for this level of formalism in closely related work. For example, the main baseline [Bagdasaryan et al.] uses a comparable amount of notation to define its setting and agents precisely.
>
> That said, while we have made every effort to only retain the minimal amount of notation for our exposition, as also explicitly appreciated by reviewers yRR8 and UWMf, if the reviewer believes it will aid readability we will further review the notation adding short intuition paragraphs after key equations. Again, we’d like to preserve rigor while making the main text as accessible as possible.
>
> - **On the presentation of the auditor and the role of adversary**
>
> We appreciate the recognition of these core contributions of our work. Please note that *the auditor is inherently defined with respect to an interaction protocol between an application agent and a third party user (in our case, an adversary)*. For this reason, we first specify what such an interaction looks like and which parties and information flows are involved, and only then introduce the auditor that evaluates leakage within this setting. Without a precise description of the interaction and the agents participating in it, the auditor itself would be underspecified.
>
> Additionally, we provide results on ASR and task completion both with and without an adversary: without an adversary (i.e. when the third party user is benign), we show that there are no leakages (see Figure 2 (b) and Figure 9) and perfect task completion for the insurance claim scenario and 95% task completion in the interview scheduling scenario, showing that the application agents function as intended without an adversary, but an adversary succeeds in breaking their privacy guardrails.
>
> We also wish to emphasize that an empirical audit, by design, evaluates an application agent against the strongest adversary one can construct. Therefore, we carefully constructed state-of-the-art adversaries to audit leakages from application agents to the best of our abilities, and spend time discussing what makes these adversaries effective (viz. looking at patterns of temporal vulnerability when using a reactive adversary vs. a longer horizon subgoals-based adversary (Sec 4.1), what successful strategies are commonly adopted by adversaries (Sec 4.4)).
>
> ---
> We hope these clarifications make it clearer how the metrics are intended to be used and how the framework is structured conceptually. We also hope it shows that the issues you raise do not reflect fundamental problems with the core contributions (the definition of implicit leakage, the CMPL audit, and the planning-based adversary), but are fully addressable (and in fact we already took action to do so) through slight expositional changes.
> Please let us know if we can provide further details.

---

> ### Author Response · Authors · 2025-11-27
> **Follow-up and clarification**
>
> Dear **Reviewer WcPM**,
>
> As the discussion period is nearing its end, we wanted to ask if there are any follow-up points we can clarify.
>
> We believe we have responded to all questions and concerns raised, in addition to taking a few days to run the additional experiments necessary to demonstrate our points, all of which are incorporated in the revised version of the paper and which we invite you to check.
> \
> If there are no further points of clarification regarding the technical nature and soundness of the work, and since the outstanding points raised were related to presentation, we kindly ask that reviewer WcPM consider increasing their score to reflect the improvements and clarifications we have provided. We are happy to continue to engage in discussion and answer any questions!

---

### Official Review · Reviewer_UWMf · 2025-11-01

**Soundness:** 3
**Presentation:** 3
**Contribution:** 4
**Rating:** 6
**Confidence:** 4

**Summary:**

This paper studies attacks on LLM agents that attempt to steal private information. In this setting the leakage is "contextual" ie an agent needs to decide for each context what can and cannot be shared. The paper introduces an auditor framework to measure and test the leakage with formal setup. Additionally, the paper sets up both long-horizon adversaries and multi-turn scenarios.

**Strengths:**

- Threat model with multi-turn adversaries is novel and has not been studied before
- This setup requires sophisticated auditor and tracking the leakage across turns.
- Paper identifies different strategies by the adversaries.
- Formal definitions for leakages

**Weaknesses:**

- Auditor uses LLM for determining leakage opens up to missing leakage or masked leakage
- A victim LLM might be sharing much more data than Auditor can catch, for example answers like Yes/No might be hard to spot when the question to the victim is obfuscated for the auditor
- Scenarios could be extended to different settings, i.e. trading, appointments, etc
- I would like to have a more substantial discussion on defenses

**Questions:**

Addressing questions above and adding additional experiments would significantly improve the paper.

---

> ### Author Response · Authors · 2025-11-20
> **Rebuttal to Reviewer UWMf (Part 1 of 2)**
>
> Thank you for your positive assessment of our work. We are glad you find our threat model _formal_ and _novel_. We address the stated weaknesses below.
>
> **Weakness 1: Auditor uses LLM for determining leakage opens up to missing leakage or masked leakage.**
>
> Thank you for this question. This concern may be based on a slight misunderstanding of our auditor's role and our paper's results. Let us elaborate:
>
> * First let us clarify the auditing role. *An auditing role is distinct from a jailbreak in the wild* and we specifically stress that *the auditor is not an active defense/guardrail*.
> \
> During an audit, the auditor goal is to observe the vulnerability of a model post-hoc, (not to intervene or act as a guardrail). **The auditor analyzes conversation logs and has no influence over/participation in the conversation.** The model is ran offline, and does not take place in an online (deployed) setting.
> This is a standard auditing assumption where the auditor has full knowledge (including the adversary's prompts) to assess if a violation occurred.
> Why? Because the system being audited should be tested aganist the strongest possible attacks and contingencies and the auditor needs to distinguish true positives from false negatives (thus it needs to have access to the "true label").
> * Next, please notice that  **the adversary's questions are not obfuscated.** As mentioned above, the auditor is an offline, post-hoc evaluator with full knowledge of the conversation history, target attribute value, etc. Thus, the adversary's questions are *not* obfuscated to the auditor. In turn its job is to flag leakages based on the whole conversation history. There is no risk of the auditor not being able to see the adversary's questions or any obfuscation and providing an insufficient evaluation.
> * Additionally, the auditor's reliability is **extensively studied in Section 4.3**. We show that, in the extensive cases analyzed, for explicit leakage, our LLM-judge auditor achieves **perfect precision (0% False Positives)** and **near-perfect recall** across all scenarios, *despite* the adversary's stealthy, multi-turn questioning.
> \
> Let us also share that our choice for an LLM-based auditor (for explicit leakage) was not rushed. We too were reluctant using LLM-based evaluators at first, and explored extensively possible alternatives, ranging from key-word match to BLEU score based auditing.  But after months of research, we concluded this design was indeed the strongest, drastically outperforming these alternatively robustly. We will be happy to include a section regarding this topic if the reviewer finds it useful.
> These results show that "missing leakages" are exceedingly rare. Please note that we also tested its robustness across different base models (Appendix H.1.6), finding high agreement.
> * Finally, note that the implicit leakage auditor is *not* LLM-based and is deterministic, and yields perfect leakage detection, being impervious to the concerns raised about LLMs-as-judges.
>
> **Weakness 2: Scenarios could be extended.**
>
> | **Credit Card Application Scenario: Target** | **ASR (Subgoals-Based)** | **ASR (Reactive)** |
> |----------------------------------------------|:------------------------:|--------------------|
> | Criminal History                             |            15            |         30         |
> | Past Insurance Claims                        |            65            |         65         |
>
> We agree with the concern about generalizability and note that our paper already includes an additional, distinct scenario in the appendix: **Credit Card Application**, which in turn considers two different targets (criminal history and insurance claims).
>
> To better emphasize this in the main text, we will highlight the key results from the Credit Card Application scenario (currently in Appendix H.1.1, Figures 7–8) in Section 4.1 of the main paper, subject to page limits. We also note that the interview scheduling scenario in the main paper explores a closely related setting to appointment scheduling.
>
> Importantly, across multiple scenarios and targets, our attack paradigm consistently outperforms baseline state-of-the-art adversaries. *Thus, the observed attack success rates and agent privacy vulnerabilities are a consequence of the methodology rather than artifacts of any particular scenario choice*.
>
> **Weakness 3: I would like to have a more substantial discussion on defenses.**
>
> We agree that a more extensive discussion of defenses strengthens the paper. In response, we have added a dedicated ``Discussion on Mitigations'' section in the revised draft (Section 4.5, highlighted in blue).
>
> (Response continued below.)

---

> > ### Author Response · Authors · 2025-11-20
> > **Rebuttal to Reviewer UWMf (Part 2 of 2)**
> >
> > **Weakness 3: I would like to have a more substantial discussion on defenses.** (Continued Response)
> >
> > |      | ASR |
> > |-----|:--:|
> > | **Without Temporal Guardrail** |  65 |
> > | **With Temporal Guardrail**    |  25 |
> >
> > * **Novel defenses motivated by our findings.** Our paper proposes new mitigation directions that are derived directly from the empirical insights in Section 4.1 (Point 3, temporal dynamics of leakage). In particular, in the revised version we have introduced a **temporal guardrails** methodology. Motivated by the observation of a peak in vulnerability around 10–20 turns (Fig. 2c), this strategy condition the agent on a simulated conversation history such that, at deployment time, it behaves more like an agent in later rounds where leakage is less likely. Additionally **we run new experiments** to test this strategy: In the insurance claim scenario targeting family medical history with a subgoals-based adversary, conditioning the agent with 40 simulated rounds of conversation reduces the ASR from $65%$ to $25%$ (see **Section 4.5** of the updated manuscript, additions in blue) without reducing benign utility (task completions in a benign setting).
> > \
> > This shows that an agent with a longer effective conversation history has a substantially reduced, though nonzero, leakage likelihood, and thus temporal guardrails provide a practical mitigation that can be combined with other defenses to further strengthen privacy protections. At the same time, the core contribution of the present work is the auditing framework itself, which we aim to study rigorously within the space limitations of this paper. A more exhaustive exploration and development of sophisticated defenses is an important direction for future work building on our framework.
> >
> > * **Auditing Leakage with SOTA Defenses:** As detailed in our response to Reviewer S6zf, we have run **new experiments** against the SOTA **dynamic data firewall** defense [Abdelnabi et al., 2025]. Please note that the AirGapAgent defense [Bagdasaryan et al., 2024], which was originally proposed for a single-turn setting, can be circumvented by prompt injections and preference manipulation, as noted by [Abdelnabi et al., 2025] (which includes the first author of [Bagdasaryan et al., 2024]).  Since privacy and security guarantees are inherently worst case, even a single successful attack suffices to deem a defense inadequate.
> > \
> > For this reason, our new experiments focus on the multi-turn dynamic firewalls of [Abdelnabi et al., 2025], which are explicitly proposed as a remedy for prompt injection vulnerabilities such as those affecting AirGapAgent.
> >
> > The table below summarizes our findings in the insurance claim scenario, reporting the attack success rate (ASR) on the intended target attribute, benign task completion rate, and the ASR on other sensitive attributes:
> >
> > |                              | **Target ASR** | **Benign Task Completion (\%)** |
> > |------------------------------|:--------------:|:------------------------:|
> > | **Without Firewalls**        |       65       |            100           |
> > | **With Defensive Firewalls** |        5       |             0            |
> >
> > However, when applying these dynamic firewalls ([Abdelnabi et al., 2025]) in the insurance claim scenario we noticed that *the data abstraction policies obtained for the data firewall (which plays a role analogous to the data minimizer in AirGapAgent) systematically recommend abstracting away all key attributes in the data, including all attributes necessary to complete the task.
> > **Thus while the defensive firewalls may defend against attacks, they also yield $0$ task successes in this scenario, making it unusable in practice**.
> > In addition, despite the authors’ claim that training on one target or data subject generalizes protection to other sensitive attributes in their setting (Sec. 4.2 in their paper), *we find instances (30% of conversations) in which other sensitive attributes (such as substance use or mental health conditions) are leaked* even when the intended target attribute (family medical history) is protected.
> >
> > Please note that we have made every effort to implement and evaluate the dynamic firewalls as faithfully and favorably as possible, testing several initializations and configurations. These new findings are presented in Section 4.5 of the main paper and Appendix H.4 (additions highlighted in blue), where we also include example logs illustrating the data firewall’s learned abstraction policies.
> >
> > ---
> >
> > Thank you again for your time and feedback. We believe that the additional results and explanations included in our response have addressed the concerns presented, but we encourage you to follow up on any points that remain unclear. Additionally, we encourage the reviewer to check the updated version of our submission, which has been extended to incorporate these additional results and discussions. We hope these revision will merit your stronger support for this work!

---

> > > ### Author Response · Authors · 2025-11-25
> > >
> > > Dear Reviewer,
> > >
> > > We thank you again for your review and engagement with our paper. As the discussion period is underway and ends in about a week, we look forward to further engaging with you to make sure all of your concerns are addressed.
> > >
> > > Looking forward to a fruitful discussion period!
> > >
> > > Sincerely,
> > >
> > > The Authors

---

### Official Review · Reviewer_yRR8 · 2025-11-02

**Soundness:** 3
**Presentation:** 3
**Contribution:** 3
**Rating:** 8
**Confidence:** 4

**Summary:**

This paper presents an audit framework for LLM agents to elicit and identify privacy leakage through multi-turn interactions with an adversarial agent. The adversarial agent converses with the target application agents to try to gain unauthorized access to sensitive information via an iterative probing strategy. The evaluation proves this method to achieve substantially higher attack success rate than the static jailbreaking methods. It also performs analysis about the temporal dynamics of leakage and strategies adopted by the adaptive adversaries. From the analysis of the audit agent, it shows the importance of auditing not only explicit leakage and also the implicit leakage that can be inferred through a series of observations.

**Strengths:**

- This paper makes a substantial contribution by expanding the investigation of privacy threats in LLM agents from a single disclosure event to a dynamic, multi-turn interaction scenario. This captures a more realistic problem setup and also demonstrates a higher attack success rate, showing that the privacy vulnerabilities in these agents are even more severe than what has been revealed in prior work.
- The paper is clearly and effectively written. The evaluation is thorough and methodologically solid. The proposed methods are novel, and the analysis conveys many insights.

**Weaknesses:**

- Although the audit framework is useful for understanding the capabilities of the attackers, I feel the paper has a limited study and discussion of mitigations against this type of adaptive attack. Specifically, the application agent does not explore alternative designs or varied ways to safeguard information. This raises questions about whether the high attack success rate reflects the current frontier of agent privacy capabilities, or whether it is more an artifact that could be addressed with a "best-effort" application agent. Providing some guidance on possible mitigation directions also feels necessary for a paper that identifies a new type of threat.

**Questions:**

- Could you explain more about the design considerations of the application agents, and discuss potential mitigation approaches?

---

> ### Author Response · Authors · 2025-11-20
> **Rebuttal to Reviewer yRR8 (Part 1 of 2)**
>
> Thank you for your positive and constructive feedback. We are glad you found this work to be a _"substantial contribution"_ with a _"thorough and methodologically solid"_ evaluation.  We respond below to your question:
>
> **Q: Limited study and discussion of mitigations.**
>
> **A:** This is an excellent point, and we are happy to expand on this.
>
> **Regarding the strength of our safety instructions and baseline defenses:** We first clarify that our application agent is intended as a best effort instantiation of safety practices that are actually used in deployed systems. Although several stronger defenses have been proposed in the literature, they typically incur substantial overhead (for example, additional LLM calls) or introduce nontrivial utility tradeoffs (as we discuss below for state of the art defenses), and they remain vulnerable to known attacks such as prompt injection.
> \
> In practice, prominent deployers (for example, Anthropic) strongly rely on safety instructions in system prompts, together with post-training alignment methods, as a primary alignment mechanism [Anthropic, 2025]. Within this deployment-relevant paradigm, we have implemented the strongest defenses we could identify under our resource constraints. The instructions used in our work were carefully curated from existing instantiations (and their modifications) and hardened against current state of the art attacks and jailbreak benchmarks over many weeks of experimentation.
>
> As shown in Figure 2a, in the ``hard'' Interview Scheduling scenarios, both state of the art attack baselines [Bagdasaryan et al., 2024; Abdelnabi et al., 2025] achieve a 0% attack success rate against our agent, indicating that our baseline defenses are strong in this setting. In contrast, existing defensive baselines, in particular the state of the art dynamic firewalls of [Abdelnabi et al., 2025], do not include explicit privacy directives, whereas our application agents combine strong safety instructions with explicit privacy guidance. _The core finding of our paper is that these current best effort defenses are insufficient against our new class of adaptive attacks_. In the revised version, we have expanded the appendix (Appendix C.3) to describe how these effective baseline defenses were selected from a broader set of candidate safety instructions and prompt refinements, and we also add new experiments on state of the art defenses, as discussed below.
>
> * **[New Results] Experiments on SOTA Defenses:** Next, note that that the AirGapAgent defense [Bagdasaryan et al., 2024], originally proposed for a single-turn setting, has been shown to be vulnerable to prompt injections and preference manipulation, as reported by [Abdelnabi et al., 2025] (which includes the first author of [Bagdasaryan et al., 2024]). Since privacy and security guarantees are inherently worst case, _even a single successful attack suffices to deem a defense inadequate or at least provides a strong a red flag_. For this reason, our new experiments focus on the state-of-the-art multi-turn defense, namely the dynamic firewalls introduced by [Abdelnabi et al., 2025], which are explicitly proposed as a remedy for prompt injection vulnerabilities such as those affecting AirGapAgent.
>
> The table below summarizes our findings in the insurance claim scenario, reporting the attack success rate (ASR) on the target attribute, benign task completion, and the attack success rate on other sensitive attributes:
>
> |                              | **Target ASR** | **Benign Task Completion (\%)** |
> |------------------------------|:--------------:|:------------------------:|
> | **Without Firewalls**        |       65       |            100           |
> | **With Defensive Firewalls** |        5       |             0            |
>
> We note a peculiar aspect: When applying these dynamic firewalls ([Abdelnabi et al., 2025]) in the insurance claim scenario, the data abstraction policies obtained for the data firewall (which plays a role analogous to the data minimizer in AirGapAgent) systematically **recommend abstracting away all key attributes in the data**, including all attributes necessary to complete the task.
> \
> As a consequence, although the defensive firewalls significantly reduce the target ASR, they also yield 0 successful task completions in this scenario, which makes the defense impractical in deployment.
>
> In addition, despite the authors’ claim that training on one target or data subject generalizes protection to other sensitive attributes in their setting (Sec. 4.2 in their paper), *we find instances (30% of conversations) in which other sensitive attributes (such as substance use or mental health conditions) are leaked* even when the intended target attribute (family medical history) is protected.
>
> (Response continued below.)

---

> ### Author Response · Authors · 2025-11-20
> **Rebuttal to Reviewer yRR8 (Part 2 of 2)**
>
> Please note that we have made every effort to implement and evaluate the dynamic firewalls as faithfully and favorably as possible, testing several initializations and configurations. These new findings are presented in **Section 4.5** of the main paper and **Appendix H.4** (additions highlighted in blue), where we also include example logs illustrating the data firewall’s learned abstraction policies.
>
> **New Mitigation Discussion:** We also agree that an expanded discussion of mitigations strengthens the work. We have therefore added a dedicated “Discussion on Mitigations” section, which includes **novel defenses suggested by our findings.**
> \
> In **Section 4.5**, we propose new defense directions motivated by our empirical analysis. In particular, motivated by the observed peak in vulnerability around 10–20 turns (Fig. 2c), we propose a notion of "temporal guardrails", that is we condition the agent on a simulated conversation history so that, at deployment time, it behaves more like an agent in later rounds where leakage is less likely.
> \
> Notably, we executed new experiments to test this strategy: In the insurance claim scenario targeting family medical history with a subgoals-based adversary, conditioning the agent on 40 simulated rounds of conversation yields a reduction in ASR from (65%) to (25%) (see **Section 4.5** of the updated manuscript, additions in blue). This shows that an agent with a longer effective conversation history experiences a substantially reduced, though non-zero, leakage likelihood, and thus temporal guardrails provide a practical mitigation that can be combined with other defenses to further strengthen protections. At the same time, the main focus of the present work is the auditing framework itself, which we investigated rigorously within this paper. A more exhaustive exploration and development of sophisticated defenses is a natural direction for future work, and we hope that our framework will serve as a foundation for the community to build upon.
>
> ---
>
> Thank you again for your comments and feedback! We hope these clarifications strengthen the paper and are happy to discuss any remaining concerns.
>
> ---
>
> ## References
>
> [Anthropic 2025] Anthropic. Release notes: System prompts. https://docs.claude.com/en/release-notes/system-prompts

---

> > ### Author Response · Authors · 2025-11-25
> >
> > Dear Reviewer,
> >
> > We thank you again for your review and engagement with our paper. As the discussion period is underway and ends in about a week, we look forward to further engaging with you to make sure all of your concerns are addressed.
> >
> > Looking forward to a fruitful discussion period!
> >
> > Sincerely,
> >
> > The Authors

---

### Official Review · Reviewer_S6zf · 2025-11-11

**Soundness:** 2
**Presentation:** 3
**Contribution:** 2
**Rating:** 4
**Confidence:** 3

**Summary:**

The papers considers a setup where an LLM gent is interacting with an adversarial user. The goal of the agent is to complete a specific task, and the agent has access to a set of attributes that may not be revealed as per privacy directives. The paper proposes a framework called Conversational Manipulation for Privacy Leakage (CMPL) with adaptive adversary and auditor to detect leakage. Similar to several recent works, the framework utilizes ideas from Contextual Integrity (CI) [Nissenbaum 2004].

**Strengths:**

* Adversarial attacks against LLM agents to induce unintended disclosure are practically important, and have started to recently gain attention. The paper is timely.
* The paper leverages notions from contextual integrity, which provides an interesting platform for analyzing privacy in conversational settings.

**Weaknesses:**

* One of my main concerns is that the paper does not put its contributions into the context with respect to recent works on contextual privacy. [Bagdasaryan et al., 2024] is presented in the paper as a jailbreaking attack, whereas [Bagdasaryan et al., 2024] used notions from CI to propose adversarial context hijacking attacks and proposed a defense to prevent unintended leakage by LLM agents. Notions from [Bagdasaryan et al., 2024] such as privacy directives and information profile are used in the paper. Comparison with [Abdelnabi et al., 2025] is also a bit vague. The paper claims to build on the foundation of [Abdelnabi et al., 2025] without providing technical details of similarities and differences.
* Taking this point further, the paper considers [Bagdasaryan et al., 2024] and [Abdelnabi et al., 2025] as baseline attacks in its experiments. However, these works also present defenses to prevent contextual privacy violations. It is not clear why the paper does not evaluate its adversary against these baseline defenses.
* The proposed adaptive adversary communicates with the auditor (specifically, when its self-consistency score exceeds a threshold, the adversary appends a hidden note only visible to the auditor). This interaction between adversary and auditor contradicts practical settings -- it is not clear why an adversary would be compliant with an auditor, in fact, an adversary would try to fool an auditor in practice. An efficient auditing framework may even be used as a  potential  guardrail by the agent to detect any leakage and adapt its response(s) based on the detection. It will be important to provide details whether the auditor design is a conceptual framework designed just to measure privacy leakage (and thereby quantify the success of the adversarial user).

**Questions:**

- Why don't the authors evaluate the performance of their proposed adversaries when the agent uses AirGapAgent [Bagdasaryan et al., 2024] for data minimization at the LLM agent or data firewalls [Abdelnabi et al., 2025]?
- Are the cases in eq (2) mutually exclusive? It seems like explicit and implicit leakage conditions may hold together at the same time. If so, it will be helpful re-write eq (2) so that the cases are mutually exclusive.
- ASR and PLTC metrics are intuitive. Can the authors provide more intuition behind considering temporal dynamics of leakage?
- As mentioned in the Weaknesses, it will be important to discuss whether auditor design is primarily conceptual. The goal of an adversarial user in practice would be to *not* get detected by a trustworthy auditor, whereas in the paper, the auditor seems to 'communicate' with the adversary. It will be helpful if the authors can provide more details on this.
- Can the authors compare and contrast the two scenarios used in the experiments with the scenarios used in [Bagdasaryan et al., 2024] and  [Abdelnabi et al., 2025]?

---

> ### Author Response · Authors · 2025-11-20
> **Rebuttal to Reviewer S6zf (Part 1 of 2)**
>
> Thank you for your thoughtful review and, in particular, for your recognition of our contributions from both a methodological standpoint and in terms of their practicality and timeliness. We reply to your main points below.
>
> **Weakness 1: On Discussion about Baselines**
>
> Thank you for raising this point. You are correct that [Abdelnabi et al., 2025] and [Bagdasaryan et al., 2024] currently represent the state of the art for contextual privacy attacks, which is precisely why we selected them as our strongest baselines.
> While we already discussed both works in Section 1 and Appendix A, we have further strengthened and clarified this discussion in the revised manuscript. In particular, we now include an explicit discussion of the defense proposed in [Bagdasaryan et al., 2024] and provide a more comprehensive comparison outlining how our contributions differ from [Abdelnabi et al., 2025] (see Appendix A: Related Work). All changes are highlighted in blue in the updated version. We believe these revisions better contextualize prior work and further clarify the novelty and significance of our contributions.
>
> **Weakness 2: On Comparison against Baseline Defenses**
>
> While we have used these works as the strongest known attack baselines, your point about evaluating their *defenses* is well taken. We have therefore conducted additional experiments and now report our findings in the revised manuscript.
>
>
> **New Results:** Firstly, please note that the AirGapAgent defense [Bagdasaryan et al., 2024], which was originally proposed for a single-turn setting, can be circumvented by prompt injections and preference manipulation, as noted by [Abdelnabi et al., 2025] (which includes the first author of [Bagdasaryan et al., 2024]). Since privacy and security guarantees are inherently worst case, even a single successful attack indicates that the defense is insufficient. For this reason, our new experiments focus on the state-of-the-art multi-turn defense, namely the dynamic firewalls introduced by [Abdelnabi et al., 2025], which are also proposed as a remedy for prompt injection vulnerabilities such as those that affect AirGapAgent.
>
> |                              | **Target ASR** | **Benign Task Completion (\%)** |
> |------------------------------|:--------------:|:------------------------:|
> | **Without Firewalls**        |       65       |            100           |
> | **With Defensive Firewalls** |        5       |             0            |
>
> However, when applying these dynamic firewalls ([Abdelnabi et al., 2025]) in the insurance claim scenario we noticed that *the data abstraction policies obtained for the data firewall (which plays a role analogous to the data minimizer in AirGapAgent) systematically recommend abstracting away all key attributes in the data, including all attributes necessary to complete the task.
> **Thus while the defensive firewalls may defend against attacks, they also yield $0$ task successes in this scenario, making it unusable in practice**.
> In addition, despite the authors’ claim that training on one target or data subject generalizes protection to other sensitive attributes in their setting (Sec. 4.2 in their paper), *we find instances (30% of conversations) in which other sensitive attributes (such as substance use or mental health conditions) are leaked* even when the intended target attribute (family medical history) is protected.
>
> Please note that we have made every effort to implement and evaluate the dynamic firewalls as faithfully and favorably as possible, testing several initializations and configurations. These new findings are presented in Section 4.5 of the main paper and Appendix H.4 (additions highlighted in blue), where we also include example logs illustrating the data firewall’s learned abstraction policies.
>
> (Response continued below.)

---

> ### Author Response · Authors · 2025-11-20
> **Rebuttal to Reviewer S6zf (Part 2 of 2)**
>
> **Weakness 3: On the Auditor-Adversary "Communication" and Misconception about Auditing**
>
> Thank you for this question. It helps surface an important conceptual misunderstanding about the design of our framework: there is no contradiction, and the key point is to distinguish the role of an auditor in the context of jailbreaks observed in the wild from that of an online defense or guardrail. Our auditor is never an active defense mechanism.
>
> In our setting, the auditor's goal is to assess a model vulnerability post hoc, not to intervene during interaction. The auditor operates purely on logged conversations, has no influence over or participation in the dialogue, and is **not** deployed in an online system. The model is run offline for evaluation. *This follows a standard auditing assumption in security and privacy*: the auditor has full knowledge, including the adversary prompts and the ground-truth secret, in order to determine whether a violation occurred. Why? Because the system being audited should be tested aganist the strongest possible attacks and contingencies and the auditor needs to distinguish true positives from false negatives (thus it needs to have access to the "true label").
>
> The “hidden note” (Sec. 2.2) fits exactly into this auditing perspective. It is **not** a message addressed to the auditor. It is an *internal state* of the adversary *Predictor module ($P_\Theta$)*, which encodes its belief about the secret (for example, "I am 90% confident the secret is X"). The auditor, as an offline evaluator with full information, reads this internal prediction and compares it to the ground truth in order to score the success of an implicit attack.
>
> We have added a paragraph to Section 2.3 to make this distinction more explicit. Please let us know if you have any additional question because this is an important point.
>
> **Answers to Specific Questions**
>
> * **Q: Are the cases in eq (2) mutually exclusive?**
>     * A: An explicit leak is indeed a special case of an implicit one (but not vice versa), and we have added a clarification to make this explicit in the paper. Our framing is meant to distinguish the *mode* of auditing being performed: an auditor interested only in explicit leakage may apply the first condition in Eq. (2), whereas auditing for any form of leakage additionally requires the second condition. This separation does not preclude that an explicit leak, when present, will also satisfy the implicit leakage criterion.
>
> * **Q: Can you provide more intuition behind temporal dynamics?**
>     * A: Absolutely! The intuition is that **risk is not static**. Our analysis (Fig 2c) reveals a non-monotonic trend: vulnerability *peaks* after 10-20 turns and then declines. This has not been observed in previous works and suggests that a simple 5-turn audit would create a false sense of security, as the agent's greatest vulnerability period is yet to come. This also provides two interesting insights: It tell us when to be most vigilant about leakage and into how the privacy behavior/vulnerability of an agent evolves over time.
>
> * **Q: Can you compare scenarios?**
>     * A: Indeed. Our scenarios (Insurance Claim, Interview Scheduling) are directly comparable to those in [Bagdasaryan et al., 2024] and [Abdelnabi et al., 2025]. There are thematic overlaps between our scenarios and those in [Bagdasaryan et al., 2024], which our scenarios build upon. In addition, the scenarios explored in both baselines are semantically similar to our scenarios: they feature standard contextual privacy settings involving a data subject, an agent as a sender, a third-party user as a recipient, and the agent is expected to adhere to contextual privacy norms. One can perform such contextual privacy evaluations in an arbitrary number of scenarios.
>
>     Further, please note that our contribution is not just providing new scenarios (please notice also the Credit Card Application, in Appendix H), but more foundamentally the novel *auditing framework* and *stronger adversary* applied to them. We find across multiple scenarios and targets that our attack methods outperform baseline adversaries consistently, therefore *the reported attack success/agent privacy vulnerability is chiefly an artifact of the methodology, as opposed to that of the choice of the scenario*.
>
> ---
> Thank you, again, for your time and feedback. We believe that the additional results and explanations included in our response have addressed the concerns presented, but we encourage you to follow-up on any points that remain unclear. Additionally, we encourage the reviewer to check the updated version of our submission, which has been extended to incorporate these additional results and discussions. We hope these revisions will merit your strong support for this work!

---

> ### Author Response · Authors · 2025-11-25
>
> As the discussion period is nearing its end, we wanted to ask **Reviewer S6zf** if there are any follow-up points we can clarify.
>
> We believe we have responded to all questions and concerns raised, in addition to taking a few days to run the additional experiments necessary to demonstrate our points, all of which are incorporated in the revised version of the paper.
> \
> If there are no further points of clarification regarding the manuscript, we kindly ask that reviewer S6zf consider increasing their score to reflect the improvements and clarifications we have provided. We are happy to continue to engage in discussion and answer any questions!

---

### Meta-Review · Area_Chair_C9Yn · 2026-01-02

**Summary:**

Most reviewers acknowledge the novelty of the proposed perspective, which expands the investigation of privacy threats in LLM agents from a single disclosure event to a dynamic, multi-turn interaction setting.

However, several concerns remain to be addressed:
- Positioning of the work in relation to recent work;
- Technical and theoretical depth of the approach;
- Generalizability of the findings, for example to other scenarios and LLM agents;
- The need for a more comprehensive discussion of potential defense mechanisms.

**Reviewer Concerns:**

Reviewer S6zf:
- One of my main concerns is that the paper does not put its contributions into the context with respect to recent works on contextual privacy. *partially solved*
- It is not clear why the paper does not evaluate its adversary against these baseline defenses. *partially solved*
- The proposed adaptive adversary communicates with the auditor. This interaction between adversary and auditor contradicts practical settings -- it is not clear why an adversary would be compliant with an auditor, in fact, an adversary would try to fool an auditor in practice. *mostly solved*
---
Reviewer yRR8:
- Although the audit framework is useful for understanding the capabilities of the attackers, I feel the paper has a limited study and discussion of mitigations against this type of adaptive attack. *partially solved*
---
Reviewer UWMf:
- Auditor uses LLM for determining leakage opens up to missing leakage or masked leakage *mostly solved*
- A victim LLM might be sharing much more data than Auditor can catch, for example answers like Yes/No might be hard to spot when the question to the victim is obfuscated for the auditor *appears to be not answered*
- Scenarios could be extended to different settings, i.e. trading, appointments, etc *partially solved, though the results appear not to be not generalized*
- more substantial discussion on defenses *mostly solved*

---
Reviewer WcPM:
- Despite the paper’s comprehensiveness, the core contribution is not sufficiently clear. The metrics for “contextual privacy leakage” (explicit vs. implicit) are heuristic and somewhat circularly defined based on the chosen adversarial model and auditor. *partially solved*
- The length and technical density of the paper are not commensurate with its conceptual novelty. *unlikely to be solved*
- The evaluation is heavily conditioned on a specific adversarial agent implementation. *appears to be not answered*

**Reviewer Scores:**

Reviewer S6zf * is likely to keep or increase the score*.
Reviewer yRR8 *is likely to keep the score*.
Reviewer UWMf * is likely to keep the score*.
Reviewer WcPM * is likely to keep the score*.

---

### Decision · Program_Chairs · 2026-01-26

Reject